# Comparative Effects of Efavirenz and Dolutegravir on Metabolomic and Inflammatory Profiles, and Platelet Activation of People Living with HIV: A Pilot Study

**DOI:** 10.3390/v16091462

**Published:** 2024-09-14

**Authors:** Crystal G. Roux, Shayne Mason, Louise D. V. du Toit, Jan-Gert Nel, Theresa M. Rossouw, Helen C. Steel

**Affiliations:** 1Department of Immunology, Faculty of Health Sciences, University of Pretoria, Pretoria 0001, South Africa; louise.dutoit@up.ac.za (L.D.V.d.T.); theresa.rossouw@up.ac.za (T.M.R.); helen.steel@up.ac.za (H.C.S.); 2Human Metabolomics, Faculty of Natural and Agricultural Sciences, North-West University, Potchefstroom 2520, South Africa; nmr.nwu@gmail.com; 3Department of Haematology, Faculty of Health Sciences, University of Pretoria, Pretoria 0001, South Africa; jan.nel@up.ac.za

**Keywords:** cardiovascular disease, chronic inflammation, cytokines, dolutegravir, efavirenz, human immunodeficiency virus, metabolites, metainflammation, platelet activation

## Abstract

Antiretroviral therapy (ART) has reduced the mortality and morbidity associated with HIV. However, irrespective of treatment, people living with HIV remain at a higher risk of developing non-AIDS-associated diseases. In 2019, the World Health Organization recommended the transition from efavirenz (EFV)- to dolutegravir (DTG)-based ART. Data on the impact of this transition are still limited. The current study therefore investigated the metabolic profiles, cytokine inflammatory responses, and platelet activation before and after the treatment transition. Plasma samples from nine virally suppressed adults living with HIV and sixteen healthy, HIV-uninfected individuals residing in Gauteng, South Africa were compared. Metabolite and cytokine profiles, and markers associated with platelet activation, were investigated with untargeted proton magnetic resonance metabolomics, multiplex suspension bead array immunoassays, and sandwich enzyme-linked immunosorbent assays, respectively. In those individuals with normal C-reactive protein levels, the transition to a DTG-based ART regimen resulted in decreased concentrations of acetoacetic acid, creatinine, adenosine monophosphate, 1,7-dimethylxanthine, glycolic acid, 3-hydroxybutyric acid, urea, and lysine. Moreover, increased levels of formic acid, glucose, lactic acid, myo-inositol, valine, glycolic acid, and 3-hydroxybutyric acid were observed. Notably, levels of interleukin-6, platelet-derived growth factor-BB, granulocyte-macrophage colony-stimulating factor, tumor necrosis factor–alpha, soluble cluster of differentiation 40 ligand, as well as regulated on activation, normal T-cell expressed and secreted (RANTES) reached levels close to those observed in the healthy control participants. The elevated concentration of macrophage inflammatory protein-1 alpha was the only marker indicative of elevated levels of inflammation associated with DTG-based treatment. The transition from EFV- to DTG-based regimens therefore appears to be of potential benefit with metabolic and inflammatory markers, as well as those associated with cardiovascular disease and other chronic non-AIDS-related diseases, reaching levels similar to those observed in individuals not living with HIV.

## 1. Introduction

The human immunodeficiency virus (HIV) is responsible for the acquired immunodeficiency syndrome (AIDS) pandemic that has claimed the lives of millions of individuals since its discovery in 1981 [1,2]. According to the 2023 data released by the Joint United Nations Programme on HIV/AIDS, approximately 39.9 million (36.1 million–44.6 million) people are living with HIV (PLWH) worldwide, with approximately 7.7 million (7 million–8.3 million) of these infections being accounted for in the South African population [3,4]. The introduction of combination antiretroviral therapy (cART) has resulted in a decline in the number of new infections as well as the mortality associated with AIDS globally and in the South African setting. However, the improved life expectancy of PLWH has led to an increased risk of developing HIV-associated comorbidities [3].

In 2019, the World Health Organization (WHO) recommended the replacement of efavirenz (EFV), a non-nucleoside reverse transcriptase inhibitor (NNRTI), with dolutegravir (DTG), an integrase strand transfer inhibitor (INSTI) for first-line treatment of HIV [5]. The transition was implemented due to the reported superiority of DTG over EFV. A 2018 systematic review by Kanters et al., incorporating 70 trials with 33,148 participants randomized to 161 treatment arms, supported the transition to DTG-based regimens as the preferred first-line HIV treatment. These authors conclude that a regimen of DTG combined with two NRTIs is advantageous with respect to increased viral suppression rates, CD4+ T-cell count recovery, and treatment tolerability [6]. In addition, a low incidence of drug-drug interactions and a higher barrier to drug resistance have been reported [5]. Following the recommendation of the transition to DTG-based regimens, Neesgaard et al. [7] reported an increased risk of cardiovascular disease (CVD) in the first two years after switching to an INSTI. Subsequent studies have reported either a non-significant [8] or no increased risk after transitioning to an INSTI [9], although this could have been influenced by the nature of the prior regimen. Notably, a number of studies have found an association between the use of INSTIs, particularly DTG, with weight gain, often leading to obesity (particularly in those with normal or above-normal body mass indices before the initiation of treatment) [10]. Moreover, studies have indicated proportional relationships between overall weight gain and visceral adipose tissue volume, which, in turn, has been linked to an increased incidence of CVD [10,11].

Metaflammation, also known as metainflammation, is a relatively new term used to describe “a chronic low-grade inflammatory state induced by alterations in metabolism” [12]. This persistent, low-grade chronic inflammatory state, commonly associated with PLWH, regardless of successful viral suppression, can lead to endothelial damage resulting in CVDs, such as atherosclerosis [13]. In addition, the association between HIV and increased upregulation and release of inflammatory cytokines and chemokines has been investigated in several studies, as reviewed by Wang et al. [14], and, in particular, it has been found that dysregulation of cytokines/chemokines persists despite virally suppressive cART. Recently, Madzime et al. reported on the potential of DTG to lead to hyper-activation of platelets in vitro, and these authors speculate that this may lead to an increased risk of cardiometabolic comorbidities [15].

The current study investigated markers of metabolism and inflammatory response (metabolites, cytokines, chemokines, and growth factors), as well as markers associated with platelet activation among virally suppressed PLWH on EFV-based cART transitioning to DTG-based regimens. Subsequently, these biomarkers were evaluated for their ability to assist in predicting individuals at increased risk of developing non-AIDS-related diseases, such as CVD.

## 2. Materials and Methods

For this retrospective longitudinal study, whole blood samples were collected from nine PLWH attending the Tshwane District Hospital (Gauteng, South Africa) between 2020 and 2022. Due to the participants being recruited from primary healthcare clinics, metabolic and cardiovascular parameters were not routinely collected. All participants had normal blood pressure and urine dipstick test results. Weight was recorded but not height, so BMIs could not be calculated. While the time since HIV diagnosis was not known, all participants had CD4+ T-cell counts and HIV viral loads recorded.

Blood was drawn in blood collection tubes containing ethylenediaminetetraacetic acid (EDTA) at recruitment and six months following the adjustment of the treatment regimen. Participants were not asked to fast or adjust any lifestyle habits before the blood draw. Inclusion criteria included virally suppressed PLWH over 18 years of age, receiving EFV-based cART comprising tenofovir disoproxil fumarate (TDF), emtricitabine, and EFV (collectively known as TEE) and transitioning to the DTG-based regimen comprising TDF, lamivudine (3TC), and DTG (collectively known as TLD). Exclusion criteria included cluster of differentiation (CD)4+ T-cell counts below 150 cells/μL. The control group (*n* = 20) comprised healthy, HIV-uninfected individuals. Written informed consent was obtained from all participants and ethics approval was granted by the Research Ethics Committee of the Faculty of Health Sciences, University of Pretoria, South Africa (reference number 390/2021).

### 2.1. Sample Preparation

Whole blood samples, collected in five milliliter EDTA-containing vacutainer tubes (Becton, Dickinson and Company, Oxford, UK), were centrifuged at 1500× *g* for 10 min at 22 °C. The plasma layer was removed and stored at −80 °C until use. Prior to performing all assays, the samples were thawed at room temperature and gently mixed by vortex.

### 2.2. C-Reactive Protein Detection and Analysis

C-reactive protein concentrations were determined through nephelometry using the Atellica NEPH 630 system (Siemens Healthineers, Erlangen, Germany) as per the manufacturer’s instructions. Concentrations of CRP greater than 5 mg/L were considered elevated, as defined in the guidelines of the National Health Laboratory Service, South Africa.

### 2.3. Metabolite Detection and Analysis

Untargeted proton magnetic resonance (^1^H-NMR) spectroscopy was conducted at the Centre for Human Metabolomics at the North-West University (Potchefstroom, South Africa) in accordance with the standard operating procedures, adapted by Mason et al. [16]. Bruker TopSpin (Version 3.5) and Bruker AMIX (Version 3.9.14) (Bruker Corp., Billerica, MA, USA) were used in the analysis of the ^1^H-NMR spectra [16,17]. Metabolites were identified using online databases, including the Human Metabolome Data Bank (https://hmdb.ca/ [accessed: 18 August 2022]) and MetaboMiner (https://bio.tools/metabominer [accessed 18 August 2022]) [17]. Metabolites, including those associated with the EDTA-containing blood collection tube, and peaks of unknown metabolites, were excluded.

### 2.4. Cytokine Determination and Analysis

Quantification of cytokines, chemokines, and growth factors (interleukin [IL]-1β, IL-4, IL-6, IL-8, IL-10, IL-12, IL-13, IL-17, IL-1 receptor antagonist [RA], eotaxin, fibroblast growth factor [FGF] basic, granulocyte colony-stimulating factor [G-CSF], granulocyte-macrophage colony-stimulating factor [GM-CSF], interferon [IFN]-γ, tumor necrosis factor [TNF]-α, IFN-γ inducible protein [IP]-10, monocyte chemoattractant protein [MCP]-1, macrophage inflammatory protein [MIP]-1α, MIP-1β, and platelet-derived growth factor [PDGF]-BB) were performed using a bead-based Bio-Plex Human Cytokine Assay kit (Bio-Rad Laboratories, Inc., Hercules, CA, USA) following the methodology in the manufacturer’s instruction manual. Analysis was performed using a Bio-Plex Luminex^®^ 200™ Suspension Array System (Bio-Rad Laboratories, Inc.) and the concentration of each analyte was calculated using Bio-Plex Manager Software (Version 6.0). Results are presented as pg/mL.

### 2.5. Platelet Activation Marker Determination and Analysis

Enzyme-linked immunosorbent assays (ELISAs), namely the RayBio^®^ Human CD40L ELISA kit (RayBiotech, Inc., Peachtree Corners, GA, USA) and Elabscience^®^ Human regulated on activation, normal T-cell expressed and secreted (RANTES) ELISA kit (Elabscience Biotechnology Inc., Houston, TX, USA) were used to quantify soluble CD40 ligand (sCD40L) and RANTES concentrations, respectively. Both assays were conducted following the manufacturer’s instruction manuals and the optical density (OD) of each sample was determined using a PowerWaveX microplate spectrophotometer (BioTek Instruments, Inc., Winooski, VT, USA) set to a wavelength of 450 nm. A standard curve, generated from the OD of the standards of known concentration using GraphPad Prism (Version 8.0) software (GraphPad, San Diego, CA, USA), was used to determine the concentration of each marker. Results are presented as pg/mL and ng/mL for sCD40L and RANTES, respectively.

### 2.6. Statistical Analysis

MetaboAnalyst (Version 4.0) was used for the metabolomics statistical analysis to identify important ^1^H-NMR bins with the use of an ANOVA at 90% significance. Identified metabolites of interest were compared using Wilcoxon rank-sum tests between the two treatment groups, as well as between groups with elevated and normal CRP concentrations. Descriptive statistics, including the mean, median, standard deviations, interquartile range, and frequencies and proportions of the cytokines and platelet activation markers, were analyzed and managed using RStudio (RStudio Incorporated, Boston, MA, USA). Non-parametric Mann–Whitney tests and paired Wilcoxon matched-pairs signed rank tests were conducted at a 10 percent (%) significance level (i.e., a *p*-value < 0.01 was considered significant), chosen due to the small sample size.

## 3. Results

A total of nine cART-experienced PLWH were investigated. Of these, five participants had normal CRP levels of less than 5 mg/L, while four of the participants CRP levels were considered elevated (more than 5 mg/L). Females made up 67% and 60% of these groups, respectively. Of the 20 healthy control participants, four had elevated CRP levels. Moreover, 50% of the healthy control group consisted of female participants. PLWH (*n* = 9) had a median age of 46 (interquartile range [IQR] 40–53) years, while the five individuals with a CRP level below 5 mg/L had a median age of 40 (IQR 34–60) years. The median age of the 20 healthy controls, irrespective of CRP concentrations, and 16 controls with CRP levels less than 5 mg/L were 37 (IQR 33–46) and 37 (IQR 33–54) years, respectively. All participants receiving cART had CD4+ T-cell counts between 332 and 760 cells/μL, as determined using flow cytometry. All PLWH had undetectable HIV viral loads at the time of blood draw.

### 3.1. C-Reactive Protein

Elevated concentrations of C-reactive protein are considered a marker of underlying infection or inflammation. In the present study, four of the nine PLWH and four of the twenty HIV-uninfected control individuals had elevated CRP levels (at baseline or at the six-month time point) of greater than 5 mg/L. Analyses (summarized in Appendix A) of the EFV-treated group, including all participants (irrespective of CRP levels) demonstrated elevated levels of acetone (*p* = 0.0030) and glycine (*p* = 0.0341), as well as decreased concentrations of leucine (*p* = 0.0300) and valine (*p* = 0.0113), compared to the control group. This group further demonstrated non-significant differences in concentrations of IL-12 (*p* = 0.5229), TNF-α (*p* = 0.3449), and GM-CSF (*p* = 0.2833), and markedly higher concentrations of biomarkers associated with platelet activation (sCD40L: *p* = 0.0077, RANTES: *p* = 0.0077) than the control group.

After excluding those with an elevated CRP, increased levels of IL-12 (*p* = 0.0980), TNF-α (*p* = 0.0979), and GM-CSF (*p* = 0.0384) as well as in the platelet markers (sCD40L: *p* = 0.0185, RANTES: *p* = 0.0185) were seen in the EFV-treated group compared to the control group. In contrast, no differences were detected in the levels of acetone (*p* = 0.1012), glycine (*p* = 0.2488), leucine (*p* = 0.1893), and valine (*p* = 0.2448).

Comparing all participants who had transitioned to DTG (irrespective of CRP levels) with healthy control individuals demonstrated that levels of leucine (*p* = 0.0131), MIP-1α (*p* = 0.0817), sCD40L (*p* = 0.0659), and RANTES (*p* = 0.0659) were markedly increased. These differences were no longer significant when individuals with elevated CRP levels were excluded from the analysis (leucine: *p* = 0.1196, sCD40L: *p* = 0.1875, and RANTES: *p* = 0.1875). Levels of MIP-1α were significantly higher compared to the control group in both analyses (including individuals with elevated CRP [*p* = 0.0979]) and excluding the four PLWH and four healthy control individuals with elevated CRP concentrations [*p* = 0.0817].

Due to the differences observed in metabolites and markers of inflammation and platelet activation, participants with elevated CRP concentrations were excluded from all further analyses.

### 3.2. Metabolites

A total of 41 metabolite peaks were found to be of interest after eliminating 254 peaks which deviated from the pooled controls by more than 20% of the relative standard deviations when an ANOVA was performed. As mentioned above, metabolites associated with the EDTA-containing blood collection tube, and peaks of unknown metabolites were excluded. Wilcoxon rank-sum tests were performed using the remaining metabolites of interest. These metabolites amounted to 16 (with the significance levels and fold-changes summarized in Appendix A.

Figure 1 presents the metabolites of significance (*p* < 0.1000). Participants receiving the EFV-based regimen had decreased levels of acetoacetic acid (*p* = 0.0193; fold-change: 0.0987), creatinine (*p* = 0.0036; fold-change: 0.0551), lactic acid (*p* = 0.0067; fold-change: 0.0551), myo-inositol (*p* = 0.0943; fold-change: 0.2417), and urea (*p* = 0.0049; fold-change: 0.0551) compared to the healthy control group. Although not significant, valine (*p* = 0.2488; fold-change: 0.0987) and 3-hydroxybutyric acid (*p* = 0.2488; fold-change: 0.3518) levels were also lower than those observed in the control group. In contrast, levels of AMP (*p* = 0.0089; fold-change: 0.0609), 1,7-dimethylxanthine (*p* = 0.0243; fold-change: 0.1105), formic acid (*p* = 0.0007; fold-change: 0.0.0292), glucose (*p* = 0.0706; fold-change: 0.1810), and glycolic acid (*p* = 0.0067; fold-change: 0.0551) were higher compared to those found for the control group. Following paired analysis of the EFV-treated group and following the transition to the DTG-based regimen, 3-hydroxybutyric acid (*p* = 0.0625; fold-change: 0.6406) was the only metabolite of interest, with those on the DTG-based regimen having higher levels compared to those treated with the EFV-based regimen. 

Following the transition to the DTG-based regimen, plasma concentrations of creatinine (*p* = 0.0303; fold-change: 0.1772), formic acid (*p* = 0.0017; fold-change: 0.0353), glycolic acid (*p* = 0.0067; fold-change: 0.0918), urea (*p* = 0.0017; fold-change: 0.0353), and valine (*p* = 0.0667; fold-change: 0.3223) remained significantly lower compared to those observed for the control group. Although not significant, levels of acetoacetic acid (*p* = 0.4008; fold-change: 0.5301), lactic acid (*p* = 1893; fold-change: 0.3375), and myo-inositol (*p* = 0.5426; fold-change: 0.6357) remained lower compared to the control group. Likewise, AMP (*p* = 0.4008; fold-change: 0.5301), 1,7-dimethylxanthine (*p* = 0.1404; fold-change: 0.3197), 3-hydroxybuyric acid (*p* = 0.8201; fold-change: 0.8406), and glucose (*p* = 0.1012; fold-change: 0.3197) concentrations remained slightly, yet not significantly, elevated compared to those observed for the control group. The fold-change values provide an indication of the difference observed between the two treatments, whereas the peak intensity (shown in Figure 1) demonstrates the values obtained during the ^1^H-NMR analysis.

### 3.3. Inflammatory Markers

The levels of significance, median, and IQR of the 23 cytokines analyzed are summarized in Appendix A. Only six of the cytokines investigated, shown in Figure 2, were determined to be of significance.

Markedly elevated median concentrations of G-CSF (*p* = 0.0933; 68.10 [IQR 59.42–72.32]), GM-CSF (*p* = 0.0669; 2.04 [IQR 1.59–2.65]), and PDGF-BB (*p* = 0.0842; 113.11 [IQR 104.88–153.61]) were found in the plasma from individuals treated with EFV compared to the healthy control group. Although not statistically significant, expression of IL-6 (*p* = 0.1963; 1.91 [IQR 1.00–2.10]) and MIP-1α (*p* = 0.1246; 1.78 [IQR 1.36–2.19]) were also moderately elevated compared to the control group. Paired analysis indicated that the median concentrations of IL-6 (*p* = 0.0625; 1.36 [IQR 0.99–1.97]) and GM-CSF (*p* = 0.0625; 1.23 [IQR 0.88–1.75]) were lower in the latter group.

Following six months of receiving the DTG-based regimen, G-CSF (*p* = 0.0604; 71.48 [IQR 55.93–82.96]) and MIP-1α (*p* = 0.0598; 1.90 [IQR 1.68–1.96]) concentrations were elevated compared to the healthy control group. Notably, IL-6 (*p* = 0.9999; 1.17 [IQR 0.82–1.64]), GM-CSF (*p* = 0.6911; 1.32 [IQR 1.09–1.77]), and PDGF-BB (*p* = 0.8049; 54.91 [IQR 17.65–153.28]) concentrations in the participants receiving the DTG-based regimen decreased to levels similar to those observed for the control group.

### 3.4. Platelet Activation Markers

As shown in Figure 3, the median plasma levels of markers associated with platelet activation, sCD40L (*p* = 0.0383; 25.75 [IQR 19.85–55.18]) and RANTES (*p* = 0.0.0383; 515.00 [IQR 396.90–1103.60]) of participants treated with the EFV-based regimen were significantly higher compared to those observed in the control group. After transitioning to the DTG-based regimen, sCD40L (*p* = 0.8436; 18.00 [IQR 16.38–25.83]) and RANTES (*p* = 0.8436; 360.00 [IQR 327.60–516.60]) concentrations decreased to levels comparable to those of the control group.

## 4. Discussion

After transitioning from EFV- to DTG-based regimens, as recommended by the WHO, concerns emerged about the impact of the drug on weight gain and subsequent CVD risk in PLWH. The focus of the present study was to compare metabolic pathways, inflammatory responses, and markers of platelet activation in a small group of individuals virally suppressed on EFV-based cART and six months post-transition to a DTG-based treatment.

The involvement of EFV in altered energy metabolism is evident with the increased levels of AMP and glucose, as well as the reduced levels of acetoacetic acid, lactic acid, and myo-inositol. This involvement is substantiated through in vitro and mouse model studies in which EFV has been linked to mitochondrial alterations, including decreased mitochondrial membrane potential and oxygen consumption, inhibition of electron transport complex I enzymes, and elevated production of reactive oxygen species [18,19]. The decline in mitochondrial function subsequently increases AMP concentrations which activates AMP-activated protein kinase (AMPK), referred to as the “master switch of cellular bioenergetics” as it up-regulates glycolysis which, in turn, increases glucose uptake into the cells by stimulating glucose transporter-4 [20,21].

The use of glycolysis in energy production may result in the observed decrease in ketogenesis (involved in the production of ketone bodies–acetone, acetoacetic acid, and 3-hydroxybutyric acid), evident by the decreased levels in ketones (acetoacetic acid and 3-hydroxybutyric acid) as well as lactic acid in the EFV-treated group [22]. This hypothesis is, however, contradicted by the findings in those individuals receiving the DTG-based regimen in which acetoacetic acid and lactic acid remain, although not significantly, lower than that observed in the control group, leading to the possible involvement of HIV-associated factors, hormone regulation (including glucagon, cortisol, thyroid-associated hormones, and catecholamines resulting in the increased breakdown of free fatty acids), and insulin which is largely involved in the regulation of ketogenesis [23]. Therefore, an individual’s lifestyle and genetic predisposition may affect the observed results, which is of particular importance as samples in this study were not obtained in a fasting state and, as a result, the diet consumed before sampling could have affected these observations. These findings, obtained using the same ^1^H-NMR method, with participants receiving the prescribed DTG-based antiretroviral regimen, were confirmed in a study by du Toit et al. in which virally suppressed pregnant women living with HIV had lower concentrations of ketone bodies, including acetoacetic acid and 3-hydroxybutyric acid, compared to pregnant women not living with HIV [24].

Six months after the transition to DTG, glucose levels remained elevated, albeit only marginally so. The impact of DTG on glucose levels is still controversial with a recent study conducted in Ethiopia reporting a lower prevalence of hyperglycemia in EFV-treated participants (9.4%) compared to those treated with DTG (17.4%) [25]. Other studies conducted in the United States of America, Uganda, and Cameroon reported elevated levels of glucose in DTG-treated cohorts compared to those treated with EFV-based regimens [26,27]. In contrast, the ‘glucose metabolic safety’ of DTG has been demonstrated in a cohort of ART-naïve PLWH in Uganda initiated on a DTG-containing regimen [28]. Clearly, further investigations are needed to confirm or refute these findings.

Increased glycolic and formic acid levels in PLWH compared to HIV-uninfected controls, regardless of cART, implicate glyoxylate and dicarboxylate metabolism, which does not naturally take place in humans due to the absence of isocitrate lyase and malate synthase in the gut [29]. These enzymes, produced by bacteria, protists, and fungi, suggest the involvement of altered gut mucosal integrity, microbiome, and mucosa in PLWH, resulting in altered microbial translocation and chronic inflammation [30,31]. A healthy gut also maintains a balanced microbiome and mucosa to support beneficial bacteria, involved in producing and absorbing metabolites such as creatine, myo-inositol, urea, and valine [32]. Therefore, low levels of creatinine (a waste product following the breakdown of creatine in muscle), myo-inositol, and valine in PLWH (irrespective of cART treatment) compared to the controls may be a result of malabsorption caused by the altered microbiome and mucosa, or alternatively, be a result of limited access to nutritious foods [32].

Elevated 1,7-dimethylxanthine levels, possibly as a result of caffeine or theobromine (found in cocoa) intake before blood sampling, were observed, particularly in the EFV-treated group [33]. The higher levels of 1,7-dimethylxanthine may also have resulted in the observed reduction in myo-inositol levels within the EFV-treated group as the myo-inositol signaling pathway and absorption capabilities are negatively affected by 1,7-dimethylxanthine [34,35]. Six months after the transition to DTG-based cART, both 1,7-dimethylxanthine and myo-inositol levels were similar to those observed in the control group, thus supporting this theory.

Following six months of DTG-based cART, GM-CSF, and PDGF-BB levels dropped to those seen in healthy, HIV-uninfected individuals [33]. The role of PDGF-BB in cardiovascular pathogenesis, including atherosclerosis, pulmonary arterial hypertension, angiogenesis, and inflammation, as well as type 2 diabetes mellitus (T2DM) is well documented, as reviewed by Wang et al. [14]. Pang et al. have also reported that a positive correlation between serum PDGF-BB levels and coronary artery disease severity, plaque rupture, platelet activation, and thrombotic events exists [36]. PDGF-BB leads to cellular changes in the arterial wall due to the accumulation of lymphocytes and macrophages as well as smooth muscle cells. This, in turn, leads to occlusion of the vessel with consequent myocardial infarction and stroke [37].

GM-CSF supports myeloid lineage maintenance by enhancing eosinophil, megakaryocyte, and macrophage activity, influencing cell maturation by determining lineage affiliations (granulocyte, macrophage, monocyte), and promoting phagocytosis while inhibiting apoptosis [38,39].

On the other hand, elevated GM-CSF levels are linked to arthritis, osteoarthritis, inflammatory bowel disease, multiple sclerosis, aortic aneurysm, and obesity [40]. Obesity, a chronic inflammatory condition, causes metabolic dysfunction, impaired adipocyte function, and insulin resistance (T2DM) [40]. Moreover, GM-CSF is more closely associated with metabolic dysfunction through its interaction with adipose tissue macrophages than with adipose tissue levels and body weight [40]. Murine models with GM-CSF deficiencies have shown improved metabolic profiles, especially with respect to insulin sensitivity, compared to wild-type mice [40]. The normalization of GM-CSF levels following the transition to the DTG-based regimen should therefore be seen as beneficial. 

In contrast, G-CSF, involved in the production of monocytes and neutrophils, remained elevated before and after transitioning from EFV to DTG compared to the control cohort [41]. This growth factor has several additional functions, including aiding in the growth and migration of endothelial cells, reducing norepinephrine reuptake (responsible for regulating cognitive function, attention, and stress management) [41,42]. As in the case of GM-CSF, G-CSF production is influenced by a multitude of cells and factors which poses a challenge in speculating on the cause for the elevated levels of this biomarker observed in the current study.

Elevated levels of IL-6 have been reported in PLWH, even when virally suppressed, and have been associated with comorbidities of HIV such as CVDs and cancer. In addition, Borges et al. have reported that IL-6 is a strong predictor for fatal events in this population [43]. As such, the lower concentrations of IL-6 found in this study following the transition to DTG-based cART should be considered as a potentially important positive outcome of the DTG-based regimen.

There are indications that the myeloid immune cell lineage could have been affected by the treatment transition, as evident by a slight increase in MIP-1α concentrations. MIP-1α, produced by monocytes, macrophages, and dendritic cells during inflammation, is responsible for the recruitment of additional cells to the site of infection [44]. However, due to the marginal increase, the effect of the treatment cannot be linked to the observed result. Further follow-up is, however, warranted.

Platelets effectively facilitate leukocyte recruitment and adhesion, vasoconstriction, and pro-inflammatory cytokine release [45,46]. PLWH generally exhibit chronically elevated platelet activation markers regardless of viral suppression [47]. Higher levels of sCD40L and RANTES may result from HIV’s direct impact on platelets, microbial translocation, upregulated platelet activation factor (observed in virally suppressed PLWH), and cART use [47,48,49]. Consistent with this study, Davidson et al. found that EFV-based treatments are typically linked to higher systemic sCD40L levels, thus resulting in elevated platelet aggregation and degranulation [50]. Consequently, transitioning from the EFV- to DTG-based regimen appears to lower the levels of the platelet activation markers, sCD40L and RANTES, thereby potentially reducing platelet activation and subsequent inflammatory responses associated with increased risk of developing chronic diseases related to atherosclerosis [50].

Cardiovascular disease, including atherosclerosis, commonly affects treatment-experienced PLWH and is largely linked to metabolic syndromes associated with abdominal obesity, dyslipidemia, elevated blood pressure, T2DM, and proinflammatory and prothrombotic states [51,52]. Unexpectedly, the DTG-treated group did not present with any metabolic alterations that could be associated with the reported weight gain observed in individuals receiving DTG-based regimens. Interestingly, recent data suggests that DTG does not cause lipid accumulation but instead reduces adiponectin expression and secretion from adipocytes, resulting in hypoadiponectinemia (a causative factor of T2DM, CVD, and weight gain) [53,54]. In addition, the lower levels of platelet activation markers observed following the transition to DTG-based cART point toward a reduced risk of CVDs in these individuals. This potentially beneficial effect of DTG on platelet activation needs to be investigated further, given the association of this antiretroviral with obesity. 

Recent studies conducted in different populations showed that, despite the considerable weight gain in participants who transitioned from TEE to TLD or were initiated on the latter, PLWH on the DTG-based regimen had improved triglycerides, low-density lipoprotein, fasting glucose, and hemoglobin A1c concentrations compared to those on the EFV-based regimen [55,56,57].

Unfortunately, the small sample size of the present study reduced the robustness of the reported findings. The use of 10% significance levels in the study affects the conclusions, especially those pertaining to small changes versus large variations in the data set. Therefore, confounding factors including tobacco use, alcohol consumption, exercise regime, as well as diet may influence the results of the current study and should be accounted for in future work. However, a major strength of the study is its longitudinal nature which allowed for the evaluation of the same patients after transitioning to a different regimen. In addition, all PLWH were virally suppressed, thereby excluding the potential confounding effect of active viral replication. As such, the results give valuable insight into the effects of DTG-based cART on the metabolomic and inflammatory profiles, and the effect of this antiretroviral on platelet activation, and warrant further investigation in a considerably larger group of individuals receiving DTG-based cART.

Another limitation of this study is that the effects of TDF, emtricitabine, and lamivudine (3TC) on the biomarkers investigated are not known. The effect of each individual’s dietary intake on metabolite profiles was not accounted for in this study and, therefore, could affect the results obtained, especially considering the detection of 1,7-dimethylxanthine (a caffeine metabolite) by untargeted ^1^H-NMR analysis [33]. Future studies focused on metabolomic analyses of intracellular contents as well as microbiome metabolites, comparing triglycerides and cholesterol concentrations, may further elucidate the metabolic pathways affected in individuals receiving cART [58].

## 5. Conclusions

Although in the setting of a pilot study, the present study has suggested that the transition from an EFV- to a DTG-based regimen results in improved energy metabolism. The retained differences in creatinine, formic acid, glycolic acid, and urea may be indicative of microbiome and microbial translocation differences associated with HIV. Despite the higher MIP-1α concentration, indicative of persistent inflammation, associated with DTG-based regimens, lower circulating concentrations of IL-6, PDGF-BB, GM-CSF, sCD40L, and RANTES point to decreased inflammatory responses and platelet activation as well as reduced chronic inflammation. This, in turn, alludes to a reduced risk of CVD and other chronic non-AIDS-related diseases in individuals receiving DTG-based cART regimens. However, longitudinal studies with larger cohorts are required with the same methodology and similar design to strengthen these findings.

## Figures and Tables

**Figure 1 viruses-16-01462-f001:**
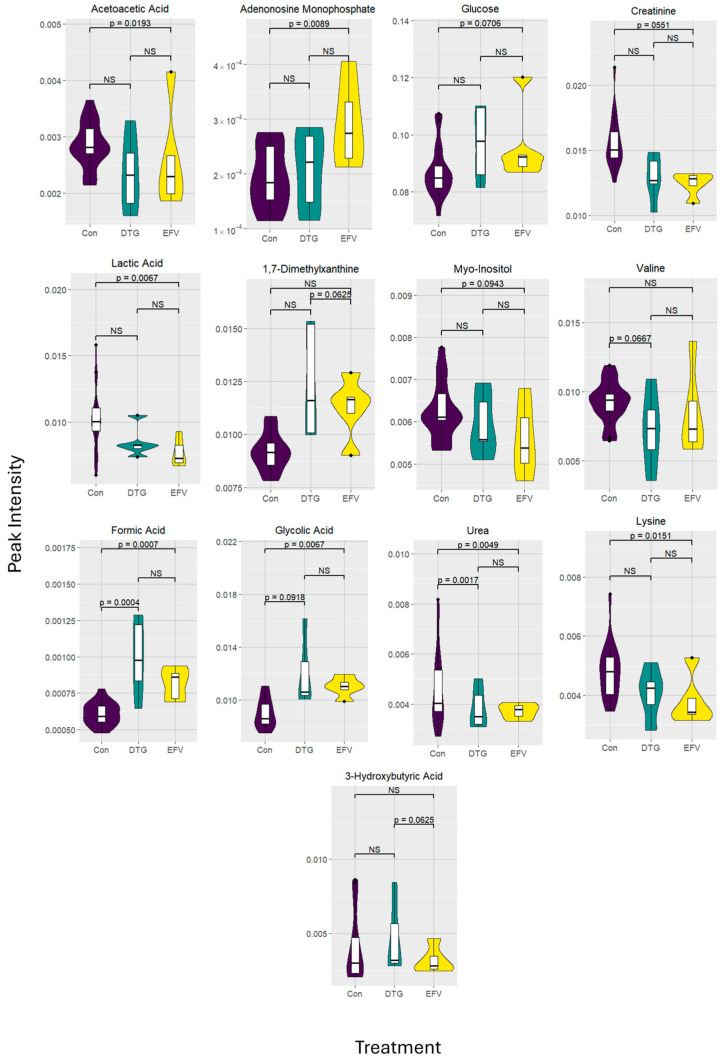
Violin and box plots depicting metabolite peak intensities of interest of five people living with HIV, before and after transitioning from an efavirenz- to a dolutegravir-based regimen, compared to sixteen healthy, HIV-uninfected control individuals. The results presented exclude participants with CRP concentrations above 5 mg/L. Levels of significance (*p*-values) between the treatment groups and control cohort are unpaired; *p*-values indicated between the two treatment groups are paired analyses. Abbreviations: Con: control; DTG: dolutegravir; EFV: efavirenz; NS: not significant.

**Figure 2 viruses-16-01462-f002:**
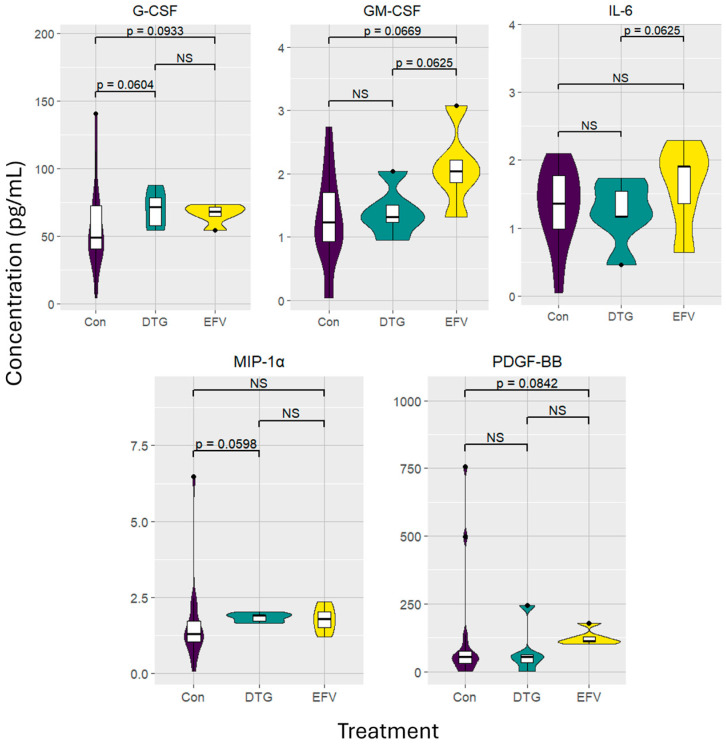
Violin and box plots depicting inflammatory markers of interest. Concentrations are depicted in pg/mL in five people living with HIV before and after transitioning from an efavirenz- to a dolutegravir-based regimen, compared to sixteen healthy, HIV-uninfected control individuals. Results exclude participants with concentrations of CRP above 5 mg/L. Levels of significance (*p*-values) between the treatment groups and control cohorts are unpaired; *p*-values between the two treatment groups are paired analyses. Abbreviations: α: alpha; Con: control; DTG: dolutegravir; EFV: efavirenz; IL: interleukin; MIP: macrophage inflammatory protein; NS: not significant; G-CSF: granulocyte colony-stimulating factor; GM-CSF: granulocyte–macrophage colony-stimulating factor; PDGF: platelet-derived growth factor.

**Figure 3 viruses-16-01462-f003:**
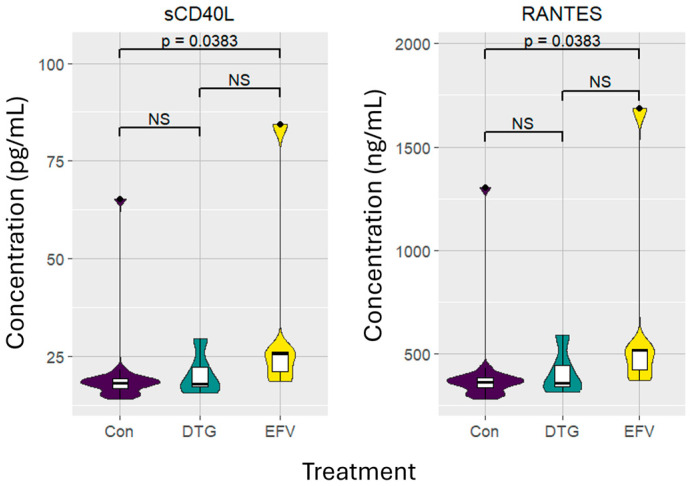
Violin and box plots of the platelet activation markers in five people living with HIV receiving efavirenz- or dolutegravir-based regimens, as well as the concentrations of sixteen healthy, HIV-uninfected controls. Results exclude participants with CRP concentrations above 5 mg/L. Levels of significance (*p*-values) between the treatment groups and control cohort are unpaired; *p*-values between the two treatment groups are paired analyses. Abbreviations: Con: control, DTG: dolutegravir; EFV: efavirenz; NS: not significant; RANTES: regulated on activation, normal T-cell expressed and secreted; sCD40L: soluble cluster of differentiation 40 ligand.

## Data Availability

Data will be made available on request.

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
