# Peer review of "Comparative Effects of Efavirenz and Dolutegravir on Metabolomic and Inflammatory Profiles, and Platelet Activation of People Living with HIV: A Pilot Study"

_viruses, 2024, doi:10.3390/v16091462_

Round 1

Reviewer 1 Report

Comments and Suggestions for Authors

In this article "Comparative effects of efavirenz and dolutegravir on metabolomic and inflammatory profiles and platelet activation in people living with HIV: a pilot study" the authors examined biomarkers including metabolites, cytokines, chemokines and growth factors, as well as markers associated with platelet activation, were measured to assess the inflammatory state of PLHIV receiving efavirenz or dolutegravir-based regimens. These biomarkers were then assessed for their ability to help predict individuals at increased risk of developing non-AIDS-related illnesses such as cardiovascular disease.

If WHO recommended such a treatment strategy, it was based on the results of some studies. It would be interesting to know if the authors' results are consistent with the studies conducted by WHO. This should be added to the article to increase the novelty of the results.

The number of people in the samples should be indicated in the figure legends, otherwise it is difficult to find. After all, these are 5 sick and 16 healthy?

Author Response

Comments 1: If WHO recommended such a treatment strategy, it was based on the results of some studies. It would be interesting to know if the authors' results are consistent with the studies conducted by WHO. This should be added to the article to increase the novelty of the results.

Response 1: Thank you for the suggestion. A section incorporating the study referenced by the WHO has been added (lines 59 to 69).

Comments 2: The number of people in the samples should be indicated in the figure legends, otherwise it is difficult to find. After all, these are 5 sick and 16 healthy?

Response 2: Thank you, the sample number of PLWH and healthy controls has been added to all figure legends.

Reviewer 2 Report

Comments and Suggestions for Authors

The study by Roux et al. analyzed the metabolic and inflammatory effects of transition from efavirenz to dolutegravir in a small cohort of PLWH participants compared to HIV-negative controls, presented as a pilot study. The authors used metabolomic analyses, bead-based multiplex and ELISA assays for inflammatory and platelet activation markers. The study is undoubtedly important and sheds light into the metabolic and inflammatory changes evident in the treatment groups. A major limitation that the authors also note, is the small sample size, particularly after exclusion of samples with CRP levels >5mg/L. Please see below for more specific comments and recommendations.

Comments:

1. Were any additional participant demographic data, metabolic or cardiovascular parameters recorded? E.g. viral load, time since HIV diagnosis/treatment initiated, body mass/BMI, insulin levels, lipid profile, blood pressure. Please include if available.

2. Lines 79-81: Please indicate if blood draws were performed fasted or not. 

3. Line 98: Please provide a reference for why 5 mg/L CRP was selected as "cutoff" for high vs. normal CRP.

4. Line 154: The included participants had CD4+ T-cell counts above 300 cells/ml. Please indicate the method and what was the range for CD4 count among the PLWH participants?

5. Line 180-182: Please clarify the sample sizes in text for the analyses that excluded CRP>5 mg/L (Control n=16; EFV and DTG n=5?)

6. Fig 1. Graphs for lactic acid, 3-hydroxybutyrate, myo-inositol and adenosine monophosphate are replicated in the fig. Why are the graphed data reported as peak intensity but data in text reported as fold-change? Please also indicate in the figure legends (here and the rest) that the CRP>5 mg/L are excluded and the n for each group.

7. Lines 273-293: Ketogenesis may be influenced by higher rates of fatty acid oxidation, insulin (or insulin resistance) and glucagon, all of which can be influenced by the metabolic state and particularly if the samples were taken fasted or not (see point 2). Please consider these factors in addition to the HIV-associated factors discussed here.

8. Lines 294-302: As above, fasted vs. non-fasted sampling may influence the glucose results, in addition to difference in treatment time between participants and, importantly, differences in the detection methods for glucose levels between this study and the cited work that found opposite results (elevated glucose with DTG) - NMR spec used here vs. other methods in the cited literature? Please include in the discussion if relevant.

9. The authors note the sample size limitation and should note the higher significance level selected (10%) limits conclusions where changes are small and variation in the data large. Please address additional confounding factors such as treatment adherence and lifestyle (smoking, alcohol consumption) as additional limitations, where relevant.

10. Perhaps emphasize in the conclusion paragraph that this pilot study warrants a more robust larger longitudinal study of similar design and methodology. 

Author Response

We would like to thank the Reviewer for the valuable input and suggestions made by them. We have addressed all the concerns raised and feel that these have added merit to the manuscript.

Comment 1: Were any additional participant demographic data, metabolic or cardiovascular parameters recorded? E.g. viral load, time since HIV diagnosis/treatment initiated, body mass/BMI, insulin levels, lipid profile, blood pressure. Please include if available.

Response 1: Thank you for the suggestion. Limited information was available for the participants of this study as the participants were recruited from primary healthcare clinics where metabolic and cardiovascular parameters are not routinely collected. All participants had a normal blood pressure and urine dipstick test. Weight was recorded but not height, so the BMI could not be calculated. While time since HIV diagnosis was not known, all participants had CD4+ T-cell counts and HIV viral loads recorded. This information has been added to the manuscript (lines 96 to 100).

Comment 2: Lines 79-81: Please indicate if blood draws were performed fasted or not.

Response 2: Thank you for drawing our attention to this point. We have stipulated that the participants were not asked to fast before the blood draw. We concede that this may have influenced the metabolomic profile observed and have added this comment in the manuscript (line 103).

Comment 3: Line 98: Please provide a reference for why 5 mg/L CRP was selected as "cutoff" for high vs. normal CRP.

Response 3: Thank you for pointing this out. The ‘cutoff’ was set at 5 mg/L as defined in the guidelines of the National Health Laboratory Service, South Africa. This has been included in the manuscript (line 123).

Comment 4: Line 154: The included participants had CD4+ T-cell counts above 300 cells/ml. Please indicate the method and what was the range for CD4 count among the PLWH participants?

Response 4: The CD4+ T-cell count range has been included in the manuscript and a comment has been added indicating that the CD4+ T-cell counts were obtained using flow cytometry (line 185).

Comment 5: Line 180-182: Please clarify the sample sizes in text for the analyses that excluded CRP>5 mg/L (Control n=16; EFV and DTG n=5?).

Response 5: Thank you for the suggestion. The sample size used for the analyses, that excluded participants with a CRP >5 mg/L, has been incorporated (line 211).

Comment 6: Fig 1. Graphs for lactic acid, 3-hydroxybutyrate, myo-inositol and adenosine monophosphate are replicated in the fig. Why are the graphed data reported as peak intensity but data in text reported as fold-change? Please also indicate in the figure legends (here and the rest) that the CRP>5 mg/L are excluded and the n for each group.

Response 6: Thank you for pointing this out. We have removed the duplicate graphs and added the sample size as well as the exclusion criteria in the figure legend. We added an explanation of fold-change and peak intensities to clarify any confusion (line 248 to 251).

Comment 7: Lines 273-293: Ketogenesis may be influenced by higher rates of fatty acid oxidation, insulin (or insulin resistance) and glucagon, all of which can be influenced by the metabolic state and particularly if the samples were taken fasted or not (see point 2). Please consider these factors in addition to the HIV-associated factors discussed here.

Response 7: Thank you for drawing our attention to this important point. The effect of other potential confounding factors including hormone regulation, insulin production and diet has been added in lines 329 to 331.

Comment 8: Lines 294-302: As above, fasted vs. non-fasted sampling may influence the glucose results, in addition to difference in treatment time between participants and, importantly, differences in the detection methods for glucose levels between this study and the cited work that found opposite results (elevated glucose with DTG) - NMR spec used here vs. other methods in the cited literature? Please include in the discussion if relevant.

Response 8: Thank you for the suggestion. We emphasized that the cited literature used the same methodology as the current study and that the PLWH were receiving the same DTG-based antiretroviral regimen (line 335 and 336).

Comment 9: The authors note the sample size limitation and should note the higher significance level selected (10%) limits conclusions where changes are small and variation in the data large. Please address additional confounding factors such as treatment adherence and lifestyle (smoking, alcohol consumption) as additional limitations, where relevant.

Response 9: Thank you for the suggestion. We have added this additional limitation in lines 445 to 449).

Comment 10: Perhaps emphasize in the conclusion paragraph that this pilot study warrants a more robust larger longitudinal study of similar design and methodology.

Response 10: Thank you for the suggestion. The need for a larger longitudinal study of similar design and methodology has been added to the concluding paragraph (lines 474 to 475).

Academic editor

Comment 1: The authors should address the comments raised by the reviewers, especially the important comments and concerns raised by reviewer 2.

Response 1: All comments and concerns raised by the reviewers have been addressed.

Comment 2: Since this is a pilot study, could the article be published as a “short communication” or similar type article? Given the comments and concerns from reviewer 2, this might not warrant a full manuscript status; however, this is obviously a manuscript reporting important findings and warrants additional “full” studies.

Response 2: Although this is a pilot study due to the small cohort of participants, the study presents a large data set including metabolic and inflammatory profiles as well as biomarkers of platelet activation. It would be impossible to present these data in the space allotted for a “short communication”. In addition, we believe that the full data set presented will assist in the development of future more comprehensive studies. As such, the authors believe that there is merit to the manuscript remaining as a research article.